# Applied Machine Learning in Industry 4.0: Case-Study Research in Predictive Models for Black Carbon Emissions

**DOI:** 10.3390/s22103947

**Published:** 2022-05-23

**Authors:** Javier Rubio-Loyola, Wolph Ronald Shwagger Paul-Fils

**Affiliations:** Centre for Research and Advanced Studies (Cinvestav), Ciudad Victoria 87130, Mexico; wolph.paulfils@cinvestav.mx

**Keywords:** industrial furnaces, black carbon, machine learning, predictive models

## Abstract

Industry 4.0 constitutes a major application domain for sensor data analytics. Industrial furnaces (IFs) are complex machines made with special thermodynamic materials and technologies used in industrial production applications that require special heat treatment cycles. One of the most critical issues while operating IFs is the emission of black carbon (EoBC), which is due to a large number of factors such as the quality and amount of fuel, furnace efficiency, technology used for the process, operation practices, type of loads and other aspects related to the process conditions or mechanical properties of fluids at furnace operation. This paper presents a methodological approach to predict EoBC during the operation of IFs with the use of predictive models of machine learning (ML). We make use of a real data set with historical operation to train ML models, and through evaluation with real data we identify the most suitable approach that best fits the characteristics of the data set and implementation constraints in real production environments. The evaluation results confirm that it is possible to predict the undesirable EoBC well in advance, by means of a predictive model. To the best of our knowledge, this paper is the first approach to detail machine-learning concepts for predicting EoBC in the IF industry.

## 1. Introduction

According to [1], Industry 4.0 refers to the next generation of industrial value generation based on the comprehensive use of internet-of-things (IoT) technology and cyber-physical systems [2]. The aim is to realize self-optimizing processes, to enable improvements in productivity as well as agility and to realize novel types of services [3]. The use of IoT technology and cyberphysical systems across the industrial value chain has led to a large number of heterogeneous data comprising, for example, machine sensor data from manufacturing as well as telemetry data from product usage [4]. Extracting business insights and knowledge from these data, e.g., for predictive maintenance or manufacturing quality analyses, is one of the major challenges in Industry 4.0 [1]. In addition, green processes are very important for the implementation of green technologies in production, to achieve positive sustainability outcomes in the Industry 4.0 era [5]. Thus, Industry 4.0 constitutes a major application domain for sensor data analytics, where the goal-oriented use of data analytics techniques represents one of the critical success factors for the realization of a sustainable Industry 4.0 [1]. This paper deals with the critical nature of bridging the gap towards the realization of Industry 4.0 in industrial furnace operations through machine-learning techniques.

Industrial furnaces (IF) are industrial machines made with complex thermodynamic materials and technologies used in industrial and commercial sectors to produce cement, ceramic, steel and other materials that require specific heat treatment cycles. An IF’s temperature dynamic range depends on the material load and the production process for this material (i.e., the load). An IF’s heat cycles typically consist of a heating phase and a cooling phase. The heating phase typically starts at ambient temperature and it may take several minutes or hours (even days) to reach a maximum temperature inside the IF. Before reaching a maximum temperature, the inner temperature may raise gradually, or step-wise, depending on the production process of the load. Either progressively or step-wise, the maximum temperature might reach 2500 (ºC) or even higher for more sophisticated furnaces. After a given period at maximum temperature, the cooling phase begins. Similarly, in the cooling phase, the inner temperature may take several minutes or hours to return to ambient temperature, progressively or step-wise.

One of the most critical issues while operating industrial furnaces is the emission of black carbon (hereafter referred to as EoBC). The EoBC is the result of a large number of factors (or their combination), such as the quality and amount of fuel (or fuel mix), furnace efficiency, technology used for the process, operation practices, type of loads [6] and other aspects related to the process conditions or mechanical properties of fluids at furnace operation [7]. Black carbon (hereafter referred to as BC) is a micro particle (e.g., diameters of 2.5 μm-$PM_{2.5}$) whose exposure to human beings can be fatal, as it may cause illnesses such as cardiovascular and lung cancer following both chronic [8,9,10,11] and acute [12] exposure.

The effects of BC have been studied in the literature. Researchers discovered that chronic BC exposure can be associated with increased blood pressure [13], faster rates of decline in lung function [14], impaired cognitive function [15], and that it can increase the chances of cardiovascular, lung cancer and cardiopulmonary mortality [16,17,18].

In order to minimize the impacts of climate change, in 2016, several countries signed the Paris agreement, for which industrial countries proposed new strategies to reduce BC emissions. Recently, countries with a high level of industrial development have taken important decisions, to protect the climate, in order to reduce black carbon emissions. In China, for example, government policy, and legal and institutional efforts are being made to bridge the gap between science policy and interdisciplinary cooperation in order to improve the black carbon inventory [19]. In the southeast area (SEA), important mitigation measures are under development due to the increased emissions of BC from various types of biomass burning , which represents a major climate risk, jeopardizing ecosystems and economies due to rapid economic growth and highly populated regions [20]. Important recommendations to the public, policymakers, and environmental bodies to control BC emissions are presented in [21].

Either for public health prevention or from regulatory viewpoints, EoBC is a highly undesirable event that must be avoided while operating IFs. The causes of the EoBC during the operation of IF are manifold. Mostly, IF cycles are programmed before the commencement of the heating phase and through all the cycle, several variables affecting IF thermodynamic performance can cause the EoBC. The EoBC can be the result of one or more variables deviating from target values, or that may have fluctuations, thereby causing the lack of control of the oxygen inside the IF for a given period of time. The sharp changes (mostly unpredictable) of variables can lead to the EoBC, so that it is paramount to anticipate and predict undesirable EoBC events during the operation of IF.

During the last decade, artificial intelligence (AI) has been used in several industrial applications. With the progressive inclusion of the industry 4.0 paradigm in the world, there is the increasing interest of several companies to optimize their processes in favour of resource utilization, product quality, environmental issues, industrial safety, etc. The challenges of EoBC can be assessed using reactive and proactive approaches. In this context, a reactive approach would take actions after the EoBC might have taken place, based on a descriptive analysis of the monitored data. Instead, a proactive approach exploits historical monitored data of IF performance in order to anticipate proactive decisions to avoid the occurrence of the EoBC. In this paper, we present a methodological approach to predict the EoBC during the operation of industrial furnaces (IFs) with the use of predictive models of machine learning (ML). Our approach makes use of a real data set of the historical operation of IFs to train ML models, and evaluate the trained models in order to identify the most suitable approach that fits best to the characteristics of the data set and its potential implementation in real production environments. In our opinion, this paper is the first attempt to develop predictive models for black carbon emissions as regards industrial furnaces. In this context, the most closely related works have used predictive models to predict black carbon in geographic areas [22], to predict $NO_{2}$ emission concentration [23], and slag foaming estimation in electric furnaces [24].

## 2. Proposed Approach, Materials and Methods

In the context of industry 4.0, the application of artificial intelligence (AI) allows turning classic production environments into intelligent environments with the ability to adapt and optimize their production processes. Machine learning is a part of AI whose application allows machines to capture real-time data and analyse historical data for estimations and optimizations of operation planning [25]. In this paper, we present the application of predictive models to anticipate the occurrence of black carbon emissions during the operation of industrial furnaces so that corrective measures can be assessed before such undesirable effects may take place. The core contribution of this paper is the use of predictive models to anticipate the undesirable EoBC. Our approach is an important tool that anticipates the potential EoBC events so that IF production managers can define, in advance, proactive actions to avoid the undesirable emissions of BC. The definition of proactive actions are beyond the scope of this paper.

### 2.1. Application Scenario and Approach

The application scenario of this paper lies in the industrial-furnace production process. There are many types of IFs, each aligned to special production and process requirements. Without loss of generality, IFs are equipped with common elements, which are briefly described hereafter. The working zone (i.e., production zone) is the most common element of all IFs and it is the element where the entire production process takes place. The combustion chambers contain burners to produce heat, valves and broadcasters to regulate gas flow, and solid structures that support all the pieces of the IF. In some cases, for better control of the industrial processes, IFs could be equipped with specific air, gas and combustion management such as turbo blowers, actuators, oxygen detectors, pressure switches or gauges, orifice metering systems, regulators, oil burners, gas burners, oxygen burners, dual-fuel burners, spark ignition systems, flame failure systems, etc. [26].

The application scenario of this paper considers an array of eight industrial furnaces (IFs), whose production cycles are executed independently and asynchronously. The eight IFs are inter-connected with a central pipeline to manage the gas emissions of the IFs. The central pipeline is connected to two post-combustion chimneys where the gas emissions are treated. The combustion chimneys are equipped with a burner at the entrance of the gas flows in order to treat volatile elements before sending the emissions to open air, where, after all, EoBC events can take place.

In general terms, sensors are utilised in a variety of different roles and industries; some examples can be found at reference [27]. Several sensors are installed in the IF infrastructure to monitor key thermodynamic variables during the production process. In the IFs’ working zone, sensors are installed to monitor temperature (the set point and the reached temperatures), air, oxygen, and atmospheric pressure. In the central pipes, sensors are installed to monitor temperature (at the IF outlet and at the central incinerators) and oxygen values. In the burners, sensors are installed to monitor burner opening (1 and 2), temperature at the output of the burner, and whether the burner is controlled automatically or manually. In the incinerators, sensors are used to monitor variables such as the temperature set point of the incinerators, the temperature within the incinerators, cooling air released, and their oxygen levels. Other sensors are installed in the production zone to monitor the temperature values classified by zones (e.g., zone A, zone B, zone C), turbulence levels, pressure levels, set point values for central incinerators, and temperature values of the walls in the IFs. All variables are monitored every fraction of second during the IF cycles and the data is used to analyse the performance of the IF infrastructure. This paper deals with the critical nature of using the monitored sensor data to train machine-learning predictive models so that they can be used to predict (i.e., with probability values) the occurrence of EoBC events during the operation of the IFs. This proposed approach is described hereafter.

Figure 1 shows the proposed approach to assess predictive models for black carbon emissions, which is split into two phases: (1) model training; and (2) model deployment. The model-training phase makes use of historical data from the database that stores the key thermodynamic variables of the IF infrastructure with the aim of training machine-learning algorithms to predict EoBC events from previous experiences. The most suitable model is used in the deployment phase to enhance the monitoring system of the IF infrastructure, with a new functionality that predicts the presence of EoBC during the operation of the IF infrastructure. The remaining sections describe the details of the proposed approach.

### 2.2. Data Set and Pre-Processing

This paper describes a case study that analyzed eight months of historical sensor data of the real-world application scenario described earlier. The data set consists of log events of the IF’s production processes monitored during the eight months of performance monitoring. The first treatment given to the data set is a labelling process, which is meant to mark the events with labels that indicate presence of black carbon in the post-combustion outlets. It is worth mentioning that the labelling process requires information about the occurrence of undesirable events of emissions of black carbon during the eight-month period of the production process data set. The tagged variable added to each event in the log is binary. A tag value “0” was added to the log events registered when no emissions of black carbon occurred during the production process. A tag value equal to “1”, on the other hand, was assigned to log events registered in the presence of BC during the operation of the IF infrastructure at production. Once tagged, a basic pre-processing treatment was applied to the data set, in order to deal with null values as well as to deal with dimensionality reduction. A feature scaling process was also applied to the variables in order to transform their values in the same range (between 0 and 1), with the intention of increasing the throughput of the predictive model. The data set after the above introductory process can be described mathematically as follows:A matrix N×M, where *N* denotes the number of instances and *M* is the number of columns or variables. In this practical case study, *N* = 400,000 and M=100.The variable to predict EoBC is binary, where a value “0” corresponds to negative instances and a value “1” corresponds to positive instances.Each vector of the data set is a tuple formally expressed as $\left(x_{j},y_{j}\right)$, for all j∈[1,N], where $x_{j}\in R^{M}$ and $y_{j}\in R^{1}$. Generally, $X\in R^{N\times M}$ and $Y\in R^{N\times 1}$.

The data set includes four types of variables, namely, 41% continuous variables, 35% discrete variables, 19% binary variables and 5% are constant.

#### Pre-Processing

One of the main challenges addressed during the pre-processing phase was the need to handle null values in the data set given the nature of the application scenario, where the IFs work in isolation and asynchronously. In this context, where sensors may not always collect data, and where the data may not include relevant information, dealing with null values in the data set is mandatory to avoid further problems that may eventually result in skewed or biased predictive models.

Another challenge addressed in the pre-processing phase was the need to address the dimensionality of the data set in this practical case study with 100 attributes. This aspect is critical, based on the fact that training a model with a data set with quite large dimensionality can lead to the overfitting phenomenon [28]. Furthermore, high-dimensional data sets can also increase the process time of the algorithms. The larger the dimensionality, the longer the time required to process the data set.

The first step towards data-set dimensionality in our case study was the determination of the correlation between all features. This procedure was achieved with the use of the *Pearson’s correlation technique* [29]. This technique is a powerful tool in finding redundancy within a data set. Data redundancy occurs when the same piece of data is stored in two or more separate places, and, thus, the need to dive deep inside the data to identify and remove redundant data. However, in our case study, the dimension of the data was slightly reduced from 100 to 94 features, which was not, indeed, a relevant reduction.

The next step in this direction was the selection of the features with direct lineal correlation with the EoBC. In this regard, the linear correlation of a variable with the EoBC can be positive or negative. In general terms, the positive linear correlation between two variables is given when one variable increases or decreases its value while the other variable in the same direction also increases or decreases its value, respectively. A negative linear correlation between two variables is given when one variable increases or decreases its value while the other, contrary, variable, decreases or increases its value, respectively. In our practical context, a positive linear correlation may take place when a given variable would result in a change from 0 to 1 in the EoBC prediction variable.

In our case study, 29 out of 94 features correlate directly with the EoBC prediction variable. The 29 features were selected using redundancy analysis. The selection of those features was carried out by considering the interpretation of the correlation coefficient proposed by Kurnia [29]. The 29 features and their correlations are graphically represented with the heat map shown in Appendix A, and more concretely in Figure A1. The variables with positive correlation are mainly oxygen- and air-oriented variables. This fact led us to draw a partial conclusion that, from the data-analysis viewpoint, the oxygen- and air-oriented variables are directly linked to the undesirable EoBC event; the higher their values, the higher the risk of EoBC occurring during the process. A reverse case was recorded in predictor variables that correlate negatively; the higher their values, the less risk there is of EoBC taking place. Moreover, after the correlation assessment, the next step is defining the most suitable prediction model for the application scenario described in this paper.

### 2.3. Selected Predictive Models for Black Carbon Emissions

The nature of the model, the criteria used to classify the variables, the popularity, effectiveness, and implementation difficulty, are all aspects considered while selecting a predictive model. The most important aspect in the context of our application domain is that the predictive model should be able to predict the emission of black carbon, which is a binary event. In this regard, the models selected in our application domain must be binary models. Regarding the classification criteria, the k-nearest neighbours (k-NN) model classifies the new data points based on the similarity measure of previously stored data points, while the support vector machine (SVM) model tries to find the optimum hyperplanes to classify points in the space. The k-NN model is popular and effective while classifying points from a *d*-dimensional space. The Adaboost model is an ensemble model designed to improve the performance of the SVM model. The logistic regression (LR) model is commonly used for many binary classification tasks, because it is easier to implement and to interpret, and very efficient to train. LR can interpret model coefficients as indicators of feature importance. Like the Adaboost model, LR can handle overfitting behaviour by aggregating the regularization function into the algorithm. Taking all the above into consideration, the logistic regression (LR), k-nearest neighbours (k-NN), support vector machine (SVM) and the ensemble learning method (Adaboost with support vector machine as estimator) are the binary predictive models selected in our case study, and their overall characteristics are briefly described in this section.

It is worth mentioning that, in order to train the models described below, the 29 features directly correlated with the tagged variable were used as inputs of the four selected models. The 29 features are, in general, related to oxygen levels, air-driven variables, temperature levels and pressure levels on specific zones of the IFs. These 29 features are the key factors for predicting EoBC during IF operation. Their combinations and interactions determine the possibility of observing EoBC during the production.

#### 2.3.1. Logistic Regression

Logistic regression (LR) is a useful quantitative procedure used to solve problems where the dependent variable is dichotomic. In machine learning, LR can be defined as a predictive analysis algorithm based on probability concepts. It has been used in extensive applications, such as document classification [30], computer vision [31], natural language processes [32] and bio-informatics [33]. For high-dimensional big data applications with limited training sampling, LR is prone to overlearn from the data and, eventually, this particular behaviour can lead to overhead. In order to overcome this issue, a regularization technique can be used for a more robust classification. For binary classification, the sigmoid function that computes the probability for a particular instance to belong to one or another class is described as follows [30]:(1)$$Prob\left(b|a\right)=\frac{1}{1+e^{-(w^{T}a)}},$$
where *a* is a sample in the data set and $w^{T}$ is the transpose of the weight vector. Considering that *m* is the number of features, $a\;\in \;R^{m}$; b∈{0,1} defines classes to which every sample belongs. The optimization LR algorithm used in this case study is L-BFGS, which is an algorithm that uses gradient descent to determine the global minimum of a data distribution with low iteration cost [34]. The regularisation constant used in this case study is $C=12\times 10^{-3}$.

#### 2.3.2. k-Nearest Neighbours

The k-Nearest Neighbours (kNN) model is a widespread classification supervised model in machine learning, known as the lazy model. This approach is non-parametric, which prescribes that no previous learning is needed to classify new events. It drives the classification process via a simple and common method based on the distance between samples. Namely, a test sample is classified by firstly obtaining the class of the *k* closest samples. The majority class of these nearest samples (or nearest single sample when *k* = 1) is returned as the prediction for that test sample [35]. For the classification of multidimensional data, various algorithms are available. For our practical case study, we used the *ball tree* algorithm [36] that builds up a metric tree by organizing data according to the metric space in which the points are localized, so that points do not need to have a finite dimension or be vectors. The algorithm splits data into two clusters, each one beeing limited by a circle (2D) or a sphere (3D) called hypersphere. At each level of the tree, the children are chosen by means of the maximum distance among each other. Equation (Equation 2) shows the formula of *Minkowsky metric* or *$L_{q}$ norm*; this metric determines the distance between d-dimensional patterns. When *q* takes values 1, 2, and ∞, the corresponding distances are *Manhattan or City Block, Euclidean* and *Chebyshev*, respectively. The distance metric used for this case study is the Chebyshev distance.
(2)$$L_{q}\left(a,b\right)=(\sum_{j=1}^{d}|a_{j}-b_{j}{{|}^{q})}^{1/q},$$
where *a* and *b* are two points in a d-dimensional space.

#### 2.3.3. Support Vector Machine

The support vector machine (SVM) approach is a powerful machine-learning tool used for either regression or classification problems. The models obtained from SVM techniques tend to generalise adequately to unseen data. Linear SVM is the simplest form of SVM and it is often used in cases of binary classification to find an optimal linear separator between the two classes of data. The optimal separator is the one that prescribes the widest margin between two classes and this means that the model is generalized and that it can be used to classify unseen data. In order to deal with over-fitting, SVM uses regularization. The regularization factor of SVM is a complex parameter denoted as *C*, which determines the trade-off between choosing a large-margin classifier and the amount by which misclassified samples are tolerated. High values of *C* imply that more importance is given to minimising the amount of misclassification rather than to finding a wide margin model [37]. SVM has been used in many classification problems, such as classifying malware events in computer operations [38]. The parameters used to optimize the algorithm were: kernel=RadialBasicFunctions(RBF) and the regularisation factor $C=38\times 10^{-3}$.

#### 2.3.4. Ensemble Learning Method (Adaboost with Support Vector Machine as Estimator)

In machine learning, ensemble learning methods are used to combine more than one predictive algorithm. These approaches are useful when the data is distributed in many databases. For some applications, in order to obtain a more accurate model, it is highly recommended to use an ensemble model. Boosting is a widely used ensemble model in machine-learning applications. Let us consider a set of *n* estimators (with n>=2), the estimators n+1 will depend completely on the estimator *n*, guiding the process in the direction of misclassified instances, and so one is assigned to each of the remaining estimators until the accuracy increases and, hence, the training error is reduced.

Adaboost is a boosting algorithm that works on the weights of incorrectly classified instances; the classifier n+1 tries to adjust the weights resulting from the classifier *n* such that subsequent classifiers will focus on more difficult cases. These techniques have been used in many applications, such as traffic classification [39]. The estimator used in our practical case study is SVM, the regularization factor was set to $C=18.5\times 10^{-8}$ and the kernel used was *radial basis functions (RBF)*.

#### 2.3.5. Bias–Variance Trade-Off: Model Selection

The selection of the model is mainly a function of the nature of the data set considered. First, the training data set in our practical use case is unbalanced and for that reason we had to deal with the bias–variance trade-off. Unbalanced data sets often lead to over fitted model and, consequently, a model with high variance. Furthermore, the high-variance model would perform well on training and validation data but with unseen data [40], the performance would be poor. Therefore, in order to avoid overfitting, we used complexity regularization to address the bias–variance trade-off. The interested reader may be referred to references [41,42,43] for a detailed explanation on the bias–variance trade-off and potential model selection options (e.g., hold-out, *k*-fold cross-validation, structural risk minimization, complexity regularization, and information criteria).

### 2.4. Metrics to Evaluate Model Performance

This subsection describes briefly the metrics used to analyse the predictive models performance, namely, *accuracy, precision, recall*, the $f_{1}$-Score and the *cross-entropy loss function*.

#### 2.4.1. Accuracy

The accuracy is directly linked to the number of instances well classified by a model. This metric determines how accurately the model is implemented. High accuracy implies that the model is well fitted; however, situations such as overfitting can appear when the accuracy is very high. In order to calculate the accuracy of a model, the following expression is used:(3)$$Accuracy=\frac{TP+TN}{TP+TN+FP+FN},$$
where: TP= true positive; TN= true negative; FP= false positive; and FN= false negative. These variables are also used in the next two definitions.

#### 2.4.2. Precision

The precision indicates the ratio of the total test instances that the classifier is able to classify as positive. In other words, the ratio of instances that the model can detect as potential risk of BC emission. Equation (Equation 4) illustrates the mathematical formula of this metric:(4)$$Precision=\frac{TP}{TP+FP}$$

#### 2.4.3. Recall

Recall can be defined as the ratio of positive instances that the classifier can detect as true positive. The mathematical formula for this metric is shown in Equation (Equation 5):(5)$$Recall=\frac{TP}{TP+FN}$$

#### 2.4.4. $f_{1}$-Score

The $f_{1}$-Score is the harmonic mean between precision and recall. A high value of this metric is required for efficient classification techniques. Expression (Equation 6) shows the mathematical formula of the $f_{1}$-Score:(6)$$f_{1}-\mathrm{Score}=2*\frac{precision*recall}{precision+recall}$$

#### 2.4.5. Cross-Entropy Loss Function

The *cross-entropy loss function*, also known as logloss for binary classification, is a metric used in ML to determine how close the prediction probability is to the corresponding actual/true value (0 or 1). The more the predicted probability diverges from the actual value, the higher the logloss value and this means that the model contains some bias [40]. Otherwise, the logloss value is close to 0 and, consequently, the model has a lot of variance, which means that it is susceptible to overfitting. Expression (Equation 7) shows the formula used to calculate logloss.
(7)$$log\;loss=-\frac{1}{N}\sum_{i=1}^{N}\left(y_{i}\;logp_{i}+\left(1-y_{i}\right)\;log\left(1-p_{i}\right)\right),$$
where *N* represents the number of observations, *y* represents the class to which a given observation belongs, and *p* is the probability of an observation belonging to a given class.

## 3. Training and Validation Results

This section evaluates the performance of the predictive models for black carbon emissions. First, results for single validation are presented, which show the performance in the *accuracy, precision, recall* and $f_{1}$-Score of the predictive models. Furthermore, cross-validation assessment results are presented to evaluate the statistical significance of the models and, also, we present the training and validation performance of the models. Finally, the predictive models are validated with unseen data, for which the evaluation results are presented. Overall, the results presented in this section could be used to determine the most suitable predictive model for the practical case study presented in this paper.

### 3.1. Single Validation

The first criteria adopted to assess the implemented models was the *single validation*, which is driven by splitting the data set into two parts: training and test data, considering the Pareto rule (80% for training and 20% for test). Table 1 shows the performance in the, *accuracy, precision, recall* and $f_{1}$-Score of each model. In descending order, the most accurate model was k-NN, followed by Adaboost, LR and SVM, with accuracy values equal to 92.55%, 84.04%, 81.91%, 80.85%, respectively. Nevertheless, the most important metric to determine the suitability of the predictive models is the $f_{1}$-Score. Thereafter, the predictive model k-NN outperforms the rest with an $f_{1}$-Score equal to 92%, followed by Adaboost with (83%), LR with (81%) and, finally, SVM with a 79%$f_{1}$-Score.

### 3.2. Cross Validation

The second validation assessment considered for the predictive models subjects of this study is *cross validation* (CV). For each model, 10-fold CV was used, and the algorithms were executed 50 times for statistical significance. Figure 2 shows the box plot for the metrics *accuracy, precision, recall* and $f_{1}$-Score for each model. The mean of each metric can be seen in Table 2. In the remaining part of this sub-section, we present a formal significance analysis procedure to determine whether the differences in the results are definitely due to the algorithms’ performance.

In order to identify the exact significance test analysis to use, we first determined the distribution of the results for each metric. For this purpose, we used the Shapiro–Wilk normality test [44]. Considering a significant level of α=0.05, the normality test confirmed that each metric has a normal distribution with a *p*-value >α.

Thus, we used a parametric statistical significance test to analyse our two hypotheses: the null hypotheses ($H_{0}$), which sates that there is no statistical difference between the means of the metrics of each model, and the alternative hypotheses ($H_{1}$), which states that the means of the metrics of each model differ statistically one from another. Next, we took the one-factor ANOVA test [45] with a significance level of p<α (α=0.05). The difference in accuracy was statistically significant, at $p=6.4272\times e^{-228}$ (p<α); the difference in precision was also statistically significant, at $p=1.7825\times e^{-216}$ (p<α). Furthermore, we reject the null hypothesis after analyzing the recall values between the four models, concluding that there is a significant difference between them of $p=9.9319\times e^{-225}$ (p<α); and, finally, the $f_{1}$-Score was also statistically different among the four models, at $P=8.9558\times e^{-226}$ (p<α).

There is no significant difference when compared with the single validation; k-NN is still the most accurate, with 97.26% as the mean accuracy and 97.06% as the *f*-Score. The second most accurate is SVM (86.29% of accuracy; 85.16% of *f*-Score), the third one was Adaboost with a mean accuracy of 84.41% and *f*-Score of 82.65%, the last one was LR with 78.8% as the accuracy and 75.92% as the *f*-Score.

### 3.3. Learning Curve

This section evaluates the performance of the predictive models over experience though their learning-curve assessments. The learning curve is a technique used in ML to interpret model behaviour. Sometimes, the predictive models are prone to presenting either overfitting or underfitting. In this regard, the learning-curve technique in our case study is helpful to identify differences among the four predictive models evaluated. The learning curve shows both the training and the validation errors. For binary classification, the error metric most widely used is the *cross-entropy loss function*, also known as *log loss*. For evaluation purposes, we initiated the training size with 50 examples and then increased the size step by step, by 50 at each step. A 10-fold cross validation was considered and the log
loss was also determined at each step.

Figure 3a shows the learning curve for the k-NN model. In this model, the training loss gradually decreases from 0.20 to 0.030 when the training-set size increases. On the other hand, the validation loss starts with stable error values around 0.25 with training-set size values between 50 and 100, and it becomes unstable, rising up to 0.37 with training-set values of 150. Moreover, the validation loss decreases up to loss values of 0.13 with a training-set size equal to 325.

The learning curve for the SVM model is graphically shown in the Figure 3b. In this model, both the training loss and the validation loss increase, from 0.10 to 0.225 and from 0.226 to 0.239 for the training and validation errors, respectively, when the training-set values rise from 50 to 200. The learning curve of the boosting model is graphically shown in Figure 3c, it prescribes a model with a little bit of bias, since both the training and the validation errors decreased synchronously and, similarly, from loss values around 0.67 to 0.37, when the training-set size increases. The learning curve of the LR model graphically depicted in Figure 3d presents a similar behaviour to the boosting model. Both the training and validation losses decrease with increasing of the training-set size, from 0.55 to 0.35 for training loss, and from 0.554 to 0.360 for the validation loss. In addition, the result prescribes a model with little bias. In these two models, the overfitting problem was addressed by means of a regularization function to avoid overfitting in both models.

### 3.4. Validation with Unseen Data

Validation with unseen data is one of the most powerful tools to assess predictive-model performance and it is a source of important information to draw conclusions about the suitability of the models under evaluation. For this purpose, this section evaluates the performance of the four predictive models with unseen data, corresponding to two production cycles of the industrial furnaces (IF) of our case study; one production cycle without BC emissions (tagged as clean
cycle), and one production cycle with BC emissions at a given time of the cycle (tagged as unclean
cycle). In the context of the real data set of our case study, the unseen data subset of the clean
cycle contains 52,201 instances while the unseen data subset from the unclean
cycle includes 38,351 instances. The four models were evaluated under the same conditions and the results produced with unseen data are graphically presented in Figure 4 and Figure 5, which are analysed holistically hereafter for a conclusive analysis.

The upper part of Figure 4 shows the temperature inside IF 46 during the productive clean
cycle, of which two aspects are relevant: (1) the moment at which the IF temperature increases, namely, when the heating procedure or the production cycle starts; and (2) the interval of the maximum temperature of the IF production cycle. Similarly, the upper part of Figure 5 shows the temperature inside IF 48 during the production cycle in which EoBC took place, for which, in addition to the two aspects mentioned earlier, one additional aspect is relevant: the exact moment at which the EoBC was registered. In this regard, the time interval of the EoBC is marked with a red spot on the graph (see the upper part of Figure 5), noting that the EoBC time interval is short.

The prediction of EoBC obtained with the k-NN, SVM, Adaboost and LR models expressed in terms of probabilities for the unseen data set of the clean and unclean cycles are shown in the bottom part of Figure 4 and Figure 5, respectively.

The k-NN model is very unstable throughout the two cycles. It predicts EoBC for the majority of the instances, with probability values equal to “1” even before the IF temperature increases and, more importantly, in both the clean and unclean cycles. The SVM is more stable than the k-NN model. Nevertheless, it also predicts the presence of BC in the majority of the instances, with EoBC probability values oscillating close to “1” even before the increase in IF temperature and, also, in both the clean and unclean cycles. The two models result in prediction probabilities mostly above 60% in the two cycles. Therefore, k-NN and SVM models present a very poor performance and are demonstrated to be unreliable for the very important task of predicting the undesirable EoBC events in the case study researched in this paper.

The Adaboost and LR models produce a very similar pattern of EoBC prediction values compared between them in both cycles. The probability results of both models are lower than 60% throughout the clean
cycle. More importantly, the predictive values in the clean
cycle have a progressive tendency, predicting a decline in values after the commencement of the IF cycle. The reduction is such that, during the interval of maximum temperature, the predictive values of EoBC probability are reduced to almost 0%. On the other hand, in the unclean
cycle, both models produce a progressive tendency, predicting a continuous increase in values of EoBC probability after the commencement of the IF cycle. The increase is such that, during the interval of EoBC, the predictive values of EoBC probability reach the maximum level around 60% in both models. Even though the BC emission was observed in a short time interval of the unclean
cycle, the progressive tendency predicts a continuous increase in the probabilities of EoBC. Both models are helpful to anticipate the undesirable EoBC event in the researched case study presented in this paper.

## 4. Technical Discussion

First, regarding the most suitable model in the researched case study. From the results presented earlier, it is clear that logistic regression (LR) and Adaboost are capable of predicting the undesirable EoBC events quite a long time in advance of its occurrence. Nevertheless, despite their similar performance, LR requires a lower processing time in comparison with Adaboost. The overhead of Adaboost is produced by the boosting algorithm, which joins together weak models that work sequentially, and the processing time depends on the sum of each weak learner. Computationally-wise, the challenge in real-time prediction is to find the algorithm with a lower time cost; considering the above, LR is the best option when predicting EoBC during real-time production in our application domain.

Second, the tendencies in the values of the EoBC probabilities reported by the models after the start of the IF cycles can be exploited to police, in real time, the probabilities for EoBC events in order to take proactive actions when the probability level crosses upwards of a given threshold defined by the industrial operation managers in order to prevent EoBC events well in advance. We have taken a step forward in this direction, whereby the upper part of Figure 6 shows the output (see the monitoring gauges in Figure 6) of the practical implementation of the approach presented in this paper, which was included in the monitoring infrastructure of our case study. The monitoring infrastructure in our case study, which is an IoT- and Cassandra-based (NoSQL) cloud computing monitoring platform, was enhanced with an LR-based predictive model implementation. The enhanced platform exploits the real-time data from the sensors in the production area, to indicate in real time the probability for EoBC. The implementation has been programmed to prescribe a very
low risk of EoBC to probability values ranging from 0 to 40% (see the green area in the monitoring gauge). In our implementation, the probability values between 41 to 60 prescribe a low risk of EoBC (the yellow area) and, respectively, probability values greater than 60 prescribe a high risk of EoBC (see the red area in the monitoring gauge).

Further directions of our work include the exploitation of the predictive model towards the optimization framework graphically presented in Figure 6 [1]. The starting point is the IF infrastructure with all sensors installed to produce data of their thermodynamic performance. The intention is to enhance the analytical capabilities from a descriptive analytical focus that could identify the root cause analyses, towards a predictive analytical focus that will forecast undesirable EoBC events. The next level in this direction is prescriptive analytics, which focus on two options: (1) defining proactive operational actions to maintain the EoBC risk within acceptable levels; and (2) optimization of the training of models when new undesirable EoBC events take place and redeploying the predictive model for continuous optimization. The optimization of the predictive approach will require the population and eventual inclusion in the machine-learning process of a database of proactive and reactive actions that could be used to enhance the analytical capabilities of the process.

## 5. Conclusions

In this paper, we proposed a solution to predict the presence of black carbon during the operation of industrial furnaces via sensor monitoring and historical data analysis. Four predictive models were selected and evaluated with real world data. Formal analysis of the data confirms that two models, namely, logistic regression (LR) and Adaboost, are suitable for predicting BC emissions. Nevertheless, computational constraints are conditional to make LR the most suitable predictive model for the practical case study presented in this paper. The implementation of the LR-based approach was added as a monitoring gauge in the real-time monitoring platform for the industrial application domain of our case study. The proposed approach is computationally efficient to report in real time the probability of EoBC, so that proactive actions can be defined before the BC is released into the atmosphere. One limitation of the approach is that it does not suggest, or define, the specific proactive actions to address the potential emission of BC. This aspect is very specific to the operational conditions that include, among others, the experience of production managers, type and quality of fuel, type and quality of loads, duration of the production cycles, quality of the production, production time constraints, etc. The inclusion of these aspects in the proposed approach is part of our future work towards an advanced optimization process to develop new operation and precautions paradigms aimed at enhancing the capabilities of organizations. To the best of our knowledge, this paper is the first approach to define machine-learning concepts in predicting black carbon emissions in industrial furnaces through real data analysis, which is an aspect that has remained almost unexplored in the area. Similarly, we envision the evaluation of predictive models in other practical applications with more case studies in areas such as production time optimization, fuel and gas optimization, and industrial-furnace design optimization, among others. We hope the ideas presented in this paper will encourage researchers and practitioners to develop more predictive approaches to enhance the business value of industrial organizations, in order to take advantage of machine-learning concepts that support the materialization of Industry 4.0 concepts for a more productive, profitable and clean industry.

## Figures and Tables

**Figure 1 sensors-22-03947-f001:**
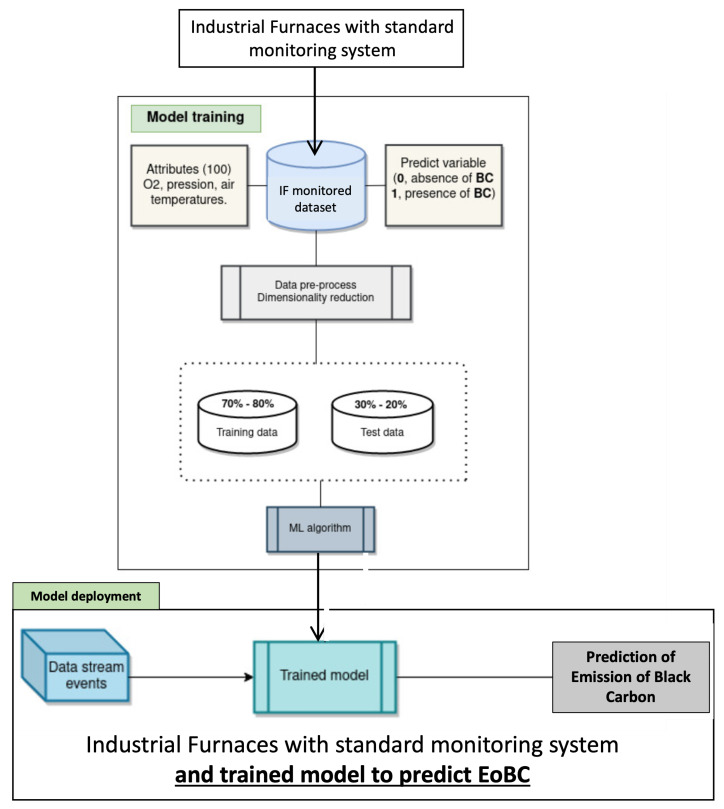
Proposed approach. Pipelines of the data process.

**Figure 2 sensors-22-03947-f002:**
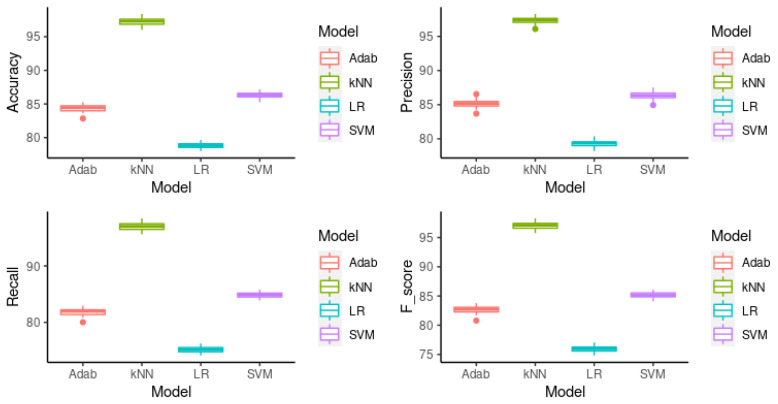
Accuracy, precision, recall, and *f*-Score resulting from 10-fold CV experiment.

**Figure 3 sensors-22-03947-f003:**
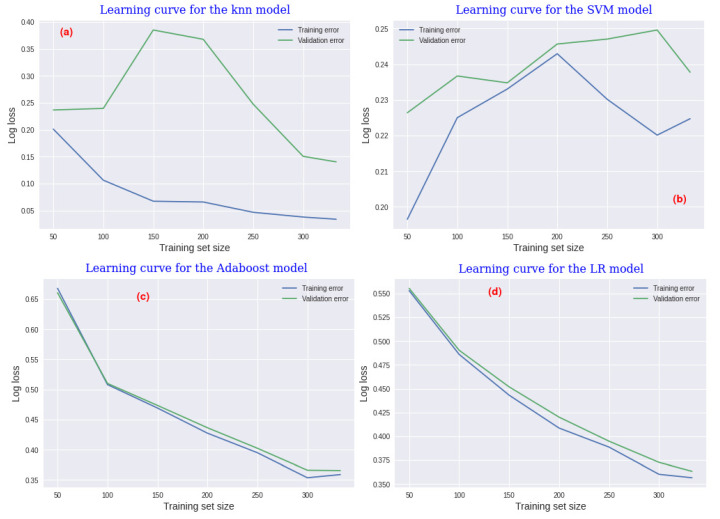
Learning curve finding by assessing each model.

**Figure 4 sensors-22-03947-f004:**
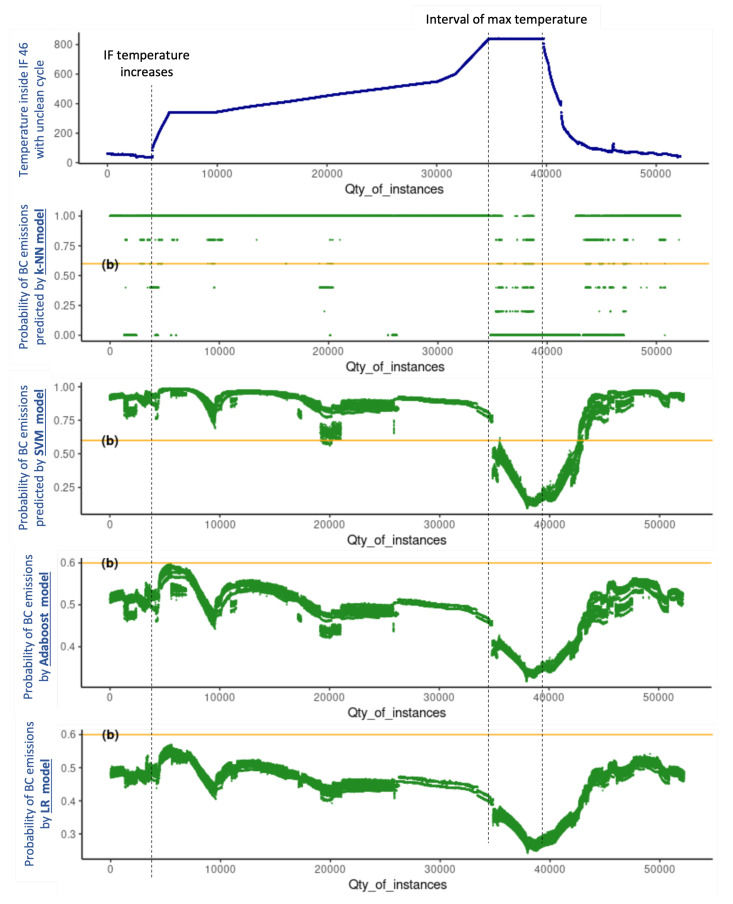
Probabilities of EoBC with clean
cycle of all models: k-NN, SVM, Adaboost and LR.

**Figure 5 sensors-22-03947-f005:**
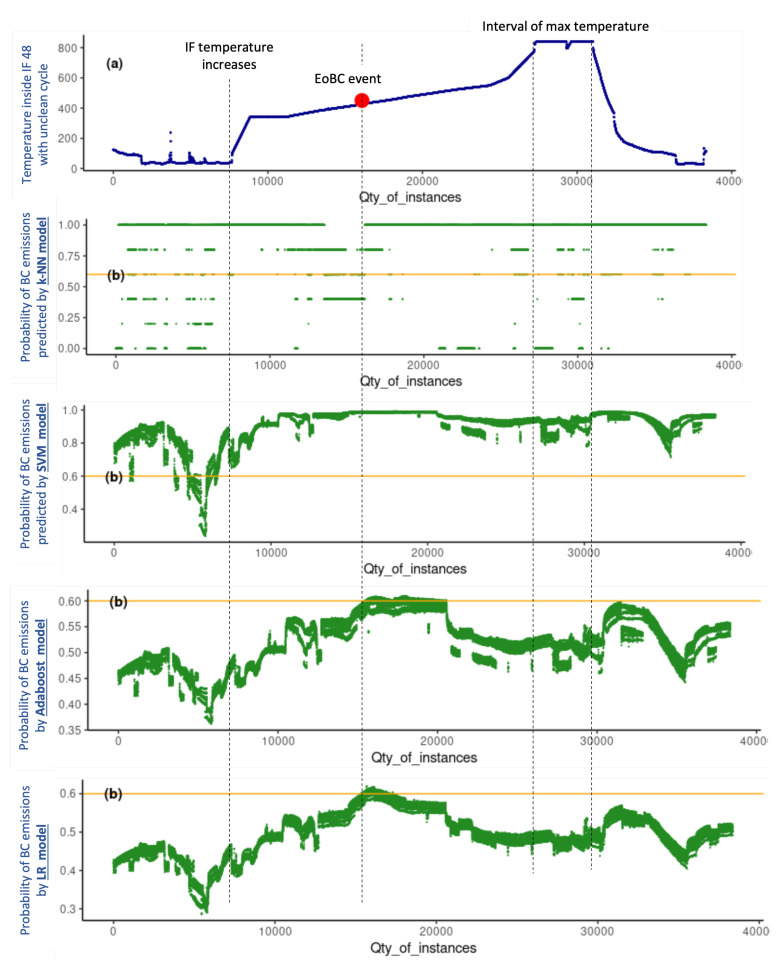
Probabilities of EoBC with unclean
cycle of all models: k-NN, SVM, Adaboost and LR.

**Figure 6 sensors-22-03947-f006:**
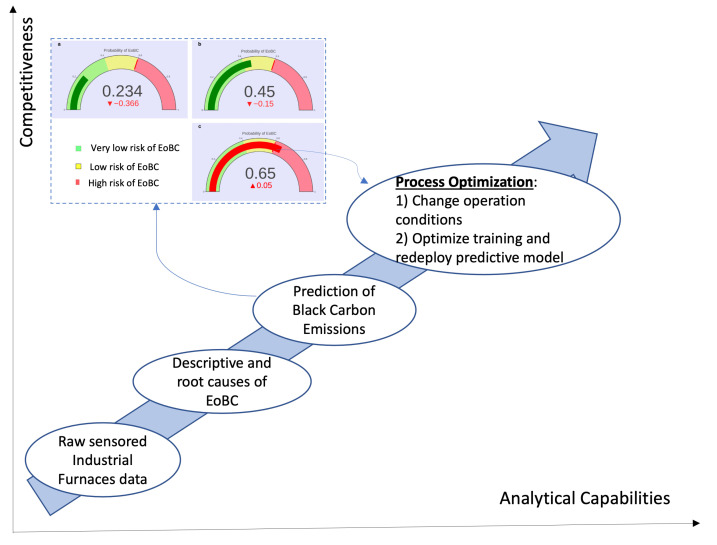
Optimization to elevate analytical capabilities and competitiveness.

**Table 1 sensors-22-03947-t001:** Results for single validation of the models.

Models	Accuracy	Precision	Recall	$f_{1}$-Score
k-NN	92.55	94.00	91.00	92.00
SVM	80.85	81.40	78.00	79.00
LR	81.91	82.00	80.00	81.00
Adaboost	84.04	83.70	82.60	83.00

**Table 2 sensors-22-03947-t002:** Result for 10-fold cross validation of the models.

Models	Mean Accuracy	Mean Precision	Mean Recall	$f_{1}$-Score
k-NN	97.26	97.35	96.98	97.07
SVM	86.29	86.33	84.86	85.16
LR	78.80	79.10	75.20	75.92
Adaboost	84.41	85.18	81.84	82.65

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
