# Peer review of "Applied Machine Learning in Industry 4.0: Case-Study Research in Predictive Models for Black Carbon Emissions"

_sensors, 2022, doi:10.3390/s22103947_

Round 1

Reviewer 1 Report

  • The linguistic quality needs some improvement. There are instances with improper/ambiguous sentences, spelling mistakes, and other writing errors. Below, only some examples are mentioned. The authors should get their paper proofread by a professional.
  1. The sentence on line 39 : “the maximum temperature may reach up to a few hundred Celsius degrees (ºC) or up to 2,500ºC, or even more for more sophisticated furnaces” looks a bit odd. Please rephrase it “the maximum temperature may reach up to 2500 (ºC) or even higher for more sophisticated furnaces”.
  2. Line 60: “manifold” instead of “manyfold” appears to be more appropriate for this sentence.
  3. Line 135: “The Figure 2” should be “Figure 2”. Same goes for all other similar instances.
  4. Line 153: “Contrary, a tag value…” is improper sentence construction.
  5. Line 369: “Followign” should be “Following”
  • The text is not properly aligned with margins. Reference [22] is out of margins.
  • A high majority of references are before 2017. There are only a few references that are after 2017. This indicates that either the authors have not done a comprehensive literature review, or the problem under study is of no more interest to the research community.
  • The text in Figure 3 is unreadable. Please increase the size of text.
  • The authors have used Logistic regression, K-NN, SVM, and Adaboost. Some motivation and discussion should be provided here as why these specific methods were chosen (an why not others from the ML domain).

Author Response

Reviewer no. 1

Comment / Recommendation 1.1 - The linguistic quality needs some improvement. There are instances of improper/ambiguous sentences, spelling mistakes, and other writing errors.

Action 1.1. The paper has been completely cleared of spelling/grammar mistakes and complete editorial work has been included in the revised paper. The paper has undergone thorough proofreading by a professional manuscript proofreading service (letter or certification attached).

Comment / Recommendation 1.2 - The text is not properly aligned with margins. Reference [22] is out of the margins.

Action 1.2. The paper has been completely formatted; references included.

Comment / Recommendation 1.3 - A high majority of references are before 2017. There are only a few references that are after 2017. This indicates that either the authors have not done a comprehensive literature review, or the problem under study is of no more interest to the research community.

Action 1.3. The emission of BC contributes to climate change, which is a major problem in the world. Recently, countries with a high level of industrial development have taken important decisions to protect against climate change to reduce black carbon emissions. Up-to-date references and text reflecting the above have been added to the revised paper (third paragraph on page 2).

Comment / Recommendation 1.4 - The text in Figure 3 is unreadable. Please increase the size of the text.

Action 1.4. The heat map has been moved to Appendix A to enhance the presentation of such a figure.

Comment / Recommendation 1.5 - The authors have used Logistic regression, K-NN, SVM, and Adaboost. Some motivation and discussion should be provided here as to why these specific methods were chosen (and why not others from the ML domain).

Action 1.5. We appreciate this comment, which led us to better position our work. In the current version of the manuscript, we have added an introductory paragraph at the beginning of Section 2.3 to justify in more detail the use of the predictive models evaluated in our research study.

Reviewer 2 Report

First of all, thank you for being able to assess such an interesting article describing "Machine Learning in Industry 4.0: A Case Study Research in Predictive Models for Black Carbon Emissions" Of course, Industry 4.0 brings with it a whole range of challenges for monitoring, processing and, above all, effective evaluation of parameters for correct decision-making. In this article, as the authors report, the creation of a predictive model for monitoring blac carbom emission has a significant impact not only in terms of thermodynamic combustion efficiency but especially in terms of environmental protection.

I have the following comments on the article:

In line 45 you describe that the EoBC is the result of a large number of factors, but when using predictive models (chapter 2.3) it is not clear what parameters enter into modeling, ie what your modeling depends on and how these parameters interact.

In chap. 2 in the line "When imminent EoBC events are predicted through our predictive approach, corrective actions should take place". On the one hand, it is not clear what events, as the combustion process and its parameters are not described, on the other hand, it is necessary to define what remedial measures are involved, as they should certainly be preventive in relation to EoBC (especially in terms of combustion efficiency)

Line 165, there is apparently a typo: (xi, yi), the index should be about j.

As I mentioned, your article is interesting but I'm not sure its applicability in practice, it compares different types of predictive models less simply. I recommend adding a description of these parameters and their impact on combustion efficiency in relation to EoBC.

Author Response

Reviewer no. 2

Comment / Recommendation 2.1 - First of all, thank you for being able to assess such an interesting article describing "Machine Learning in Industry 4.0: A Case Study Research in Predictive Models for Black Carbon Emissions" Of course, Industry 4.0 brings with it a whole range of challenges for monitoring, processing and, above all, effective evaluation of parameters for correct decision-making. In this article, as the authors report, the creation of a predictive model for monitoring black carbon emission has a significant impact not only in terms of thermodynamic combustion efficiency but especially in terms of environmental protection.

Action 2.1. We thank the reviewer for the positive comments regarding the assessment of our work and the suitability of our proposed approach in favor of the environmental benefits of Industry 4.0 approaches.

Comment / Recommendation 2.2 - In line 45 you describe that the EoBC is the result of a large number of factors, but when using predictive models (chapter 2.3) it is not clear what parameters enter into modeling, ie what your modeling depends on and how these parameters interact.

Action 2.2. We thank the referee for highlighting the lack of clarity in this respect. We have now improved the explanations provided in the paper regarding the parameters directly related to the EoBC prediction (last paragraph in Section 2.2.1).

Comment / Recommendation 2.3 - In chap. 2 in the line "When imminent EoBC events are predicted through our predictive approach, corrective actions should take place". On the one hand, it is not clear what events, as the combustion process and its parameters are not described, on the other hand, it is necessary to define what remedial measures are involved, as they should certainly be preventive in relation to EoBC (especially in terms of combustion efficiency).

Action 2.3. We thank the referee for this important observation regarding the specific proactive actions to be defined when there is a high risk of emission of BC. Our approach is an important tool that anticipates the potential EoBC events so that IF production managers can define in advance, proactive actions to avoid the undesirable emissions of BC. This aspect is very specific to the operational conditions that include among others, the experience of production managers, type and quality of fuel, type and quality of loads, duration of the production cycles, quality of the production, production time constraints, etc. We have clarified that the definition of proactive actions is beyond the scope of this paper. Moreover, we have now referenced more explicitly that the inclusion of these aspects in the proposed approach is part of our future work towards an advanced optimization process to develop new operation and precautions paradigms aimed at enhancing the capabilities of the organization.

Comment / Recommendation 2.4 - Line 165, there is apparently a typo: (xi, yi), the index should be about j.

Action 2.4. We thank the referee for this important observation regarding the index. We have corrected the typo.

Comment / Recommendation 2.5 - As I mentioned, your article is interesting but I'm not sure about its applicability in practice, it compares different types of predictive models less simply. I recommend adding a description of these parameters and their impact on combustion efficiency in relation to EoBC.

Action 2.5. We thank the referee for this important observation regarding the applicability of the approach and the description of parameters regarding the emission of BC. Regarding the applicability of the approach, the revised manuscript clarifies that implementation of the LR-based approach has been added as a monitoring gauge in the real-time monitoring platform for the industrial application domain of our case study. The proposed approach is computationally efficient to report in real-time, the probability of EoBC so that proactive actions can be defined before the BC is released into the atmosphere. One limitation of the approach is that it does not suggest, or define the specific proactive actions to address the potential emission of BC. As we mentioned earlier (Action 2.3), this aspect is very specific to the operational conditions that include among others, the experience of production managers, type and quality of fuel, type and quality of loads, duration of the production cycles, quality of the production, production time constraints, etc. The inclusion of these aspects in the proposed approach is part of our future work towards an advanced optimization process to develop new operation and precautions paradigms aimed at enhancing the capabilities of the organization. The above explanations have been added to the revised manuscript. Regarding the parameters with a direct impact on combustion efficiency in relation to EoBC, the heat map with emphasis on these variables has been moved to Appendix A to enhance the presentation of such figure.

Reviewer 3 Report

The abstract and the title of the paper are well defined.

The literature review is appropriately designed, with the authors correctly moving from general i4 to IF and then to EoBC. Here I would perhaps only recommend a paragraph on the connection between i4 and sustainability (environment): https://doi.org/10.3390/su12155968 , https://doi.org/10.3390/su12156250 , https://doi.org/10.3390/su12156007 )

Possibly also the use of sensors in terms of production, emissions, control etc., as these are a significant part of the practical part of the paper (https://doi.org/10.3390/s100605469, https://doi.org/10.3390/s90604728 )

Figure 1 should be omitted and does not add value to the paper.

Figure 3 prefers just to put it in the appendix; it is unreadable at this size.

In the text, I appreciate the clearly defined calculations with references to the literature

In the tables, it would be helpful to use two decimal places everywhere.

I recommend combining the graphs of the learning of each model in colour into one graph so that the results are more transparent. It will also make the text more attractive.

In the text's conclusion, I would appreciate mentioning the limitations of the work and possibly outlining further directions for the authors.

The paper is undoubtedly suitable for publication as it deals with the current topic of emissions and technology. One limitation is that there is only one case study.

Author Response

Reviewer no. 3

Comment / Recommendation 3.1 - The abstract and the title of the paper are well defined.

The literature review is appropriately designed, with the authors correctly moving from general i4 to IF and then to EoBC. Here I would perhaps only recommend a paragraph on the connection between i4 and sustainability (environment): https://doi.org/10.3390/su12155968, https://doi.org/10.3390/su12156250, https://doi.org/10.3390/su12156007. Possibly also the use of sensors in terms of production, emissions, control etc., as these are a significant part of the practical part of the paper (https://doi.org/10.3390/s100605469, https://doi.org/10.3390/s90604728).

Action 3.1. We thank the reviewer for the positive comments regarding our work.  We have added the corresponding text to bring the gap identified by the reviewer namely, between Industry 4.0 and sustainability, as well as emphasizing the importance of sensors in the research study presented in the manuscript.

Comment / Recommendation 3.2 - Figure 1 should be omitted and does not add value to the paper.

Action 3.2. We have removed Figure 1 as recommended by the reviewer.

Comment / Recommendation 3.3 - Figure 3 prefers just to put it in the appendix; it is unreadable at this size.

Action 3.3. Figure 3 (heat map) has been moved to Appendix A to enhance the presentation of such figure (Action 1.4).

Comment / Recommendation 3.4 - In the text, I appreciate the clearly defined calculations with references to the literature. In the tables, it would be helpful to use two decimal places everywhere.

Action 3.4. We thank the reviewer for the positive comments regarding the calculations with references to the literature. We have formatted all tables with double values.

Comment / Recommendation 3.5 - I recommend combining the graphs of the learning of each model in color into one graph so that the results are more transparent. It will also make the text more attractive.

Action 3.5. We have followed the reviewer’s recommendation to enhance the presentation of the Figure.

Comment / Recommendation 3.6 - In the text's conclusion, I would appreciate mentioning the limitations of the work and possibly outlining further directions for the authors.

Action 3.6. We thank the reviewer for pointing out the lack of clarity in the limitations and future directions of the work. We have clarified the main limitations of the approach, and have described the future directions of the work both in the last sections of the revised manuscript.

Comment / Recommendation 3.7 - The paper is undoubtedly suitable for publication as it deals with the current topic of emissions and technology. One limitation is that there is only one case study.

Action 3.7. We thank the reviewer for the positive comments about the suitability of the manuscript. We also agree with the reviewer in that the main limitation is the only case study presented. We envision the evaluation of predictive models in other practical applications with more case studies in areas like production time optimization, fuel and gas optimization, and industrial furnaces design optimization, among others. The above clarification has been included in the revised manuscript.

Round 2

Reviewer 2 Report

Thank you for submitting a modified article. My comments were accepted and added and edited in the article.